# The Application of Ejaculate-Based Shotgun Proteomics for Male Infertility Screening

**DOI:** 10.3390/biomedicines12010049

**Published:** 2023-12-24

**Authors:** Timur Shkrigunov, Victor Zgoda, Peter Klimenko, Anna Kozlova, Maria Klimenko, Andrey Lisitsa, Mark Kurtser, Natalia Petushkova

**Affiliations:** 1Laboratory of Protein Biochemistry and Pathology, Institute of Biomedical Chemistry, 119121 Moscow, Russia; lisitsa058@gmail.com (A.L.); cyp450@mail.ru (N.P.); 2Laboratory of Systems Biology, Institute of Biomedical Chemistry, 119121 Moscow, Russia; victor.zgoda@gmail.com; 3Department of Obstetrics and Gynecology, Pirogov Russian National Research Medical University, 117997 Moscow, Russia; pa.klimenko@mail.ru (P.K.); klimma@yandex.ru (M.K.); 4Center of Scientific and Practical Education, Institute of Biomedical Chemistry, 119121 Moscow, Russia; ministreliya13113@gmail.com; 5Center for Family Planning and Reproduction, Moscow Department of Health, 117209 Moscow, Russia; m.kurtser@mcclinics.ru

**Keywords:** ejaculate, spermatozoa, shotgun proteomics, testis-specific proteins, male infertility

## Abstract

Problems with the male reproductive system are of both medical and social significance. As a rule, spermatozoa and seminal plasma proteomes are investigated separately to assess sperm quality. The current study aimed to compare ejaculate proteomes with spermatozoa and seminal plasma protein profiles regarding the identification of proteins related to fertility scores. A total of 1779, 715, and 2163 proteins were identified in the ejaculate, seminal plasma, and spermatozoa, respectively. Among these datasets, 472 proteins were shared. GO enrichment analysis of the common proteins enabled us to distinguish biological processes such as single fertilization (GO:0007338), spermatid development (GO:0007286), and cell motility (GO:0048870). Among the abundant terms for GO cellular components, zona pellucida receptor complex, sperm fibrous sheath, and outer dense fiber were revealed. Overall, we identified 139 testis-specific proteins. For these proteins, PPI networks that are common in ejaculate, spermatozoa, and seminal plasma were related to the following GO biological processes: cilium movement (GO:0003341), microtubule-based movement (GO:0007018), and sperm motility (GO:0097722). For ejaculate and spermatozoa, they shared 15 common testis-specific proteins with spermatogenesis (GO:0007283) and male gamete generation (GO:0048232). Therefore, we speculated that ejaculate-based proteomics could yield new insights into the peculiar reproductive physiology and spermatozoa function of men and potentially serve as an explanation for male infertility screening.

## 1. Introduction

Despite achievements in the field of human reproductive health, there is yet to be a breakthrough in the treatment of male infertility, which affects about 23% of men of reproductive age. Meta-regression data indicate an unexplained decline in sperm (spermatozoa, the male reproductive cell, or gamete) concentrations and total sperm counts over the last four decades, suggesting an urgent need to implement further basic and clinical research in andrology [1]. However, despite that, the last World Health Organization manual concerning the standardized investigation, diagnosis, and management of male infertility was published more than two decades ago [2]. In March 2021, the WHO released the sixth edition of the laboratory manual for the examination and processing of human semen [3]. In fact, the manual consists of three parts: sperm examination, preparation, and cryopreservation of ejaculate; quality control; and assurance [4]. The progressive decline in sperm quality has lowered the spermiogram parameters proposed by the World Health Organization. Semen analysis (sperm quality examination) is featured in the sixth edition of WHO guidelines because there is increasing evidence that poor sperm quality can affect the health of offspring.

Currently, no effective drug treatment can improve the quality of a man’s sperm [5]. The most common male fertility test, which is based on sperm analysis, includes the determination of sperm concentration, motility, viability, and morphology as well as a comparison of these indicators with the reference values established by the WHO [6]. Meanwhile, identifying which semen quality indicators are comparable to those of pathozoospermia is challenging. However, clinical practice has shown that not only are the number and motility of spermatozoa important, but any inherited conditions (i.e., genetic abnormality, genetic defect, and genetic disease), chronic health problems such as heart disease, cancer, and diabetes, genitourinary tract infections, endocrine disorders, injuries, or surgeries can affect fertility. However, genetic aspects do not always have clinical manifestations, but they are extremely important for the diagnosis of male infertility in a subject [7]. OMIC technologies for the study of male infertility without overly invasive testing may help screen patients for counselling and the selection of the right therapies.

Proteomic analysis of ejaculated spermatozoa is a powerful source of information to further understand and characterize the molecular mechanisms underlying male infertility [8]. The application of bioinformatic methods for proteomic data analysis enables us to reveal the cellular pathways and protein–protein interactions (PPI) linked with normal spermatogenesis and infertility with defective spermatogenesis. Furthermore, the incorporation of diverse bioinformatic approaches can be used to identify prospective diagnostics and therapeutic targets for medication treatment of male infertility [9].

The most important step in the preparation of a semen sample for further OMIC analysis or use for in vitro fertilization is the washing of the spermatozoa to ensure the removal of seminal plasma. There are several methods that can achieve this, and each method has both advantages and disadvantages [10]. Among the shortcomings of washing procedures, one can distinguish cell damage, the non-removal of debris and leukocytes, and spermatozoa losses during several repetitive centrifugation steps. Furthermore, low values of the proportion of motile spermatozoa in samples of non-normozoospermic ejaculate are a major challenge, not only for the procedure of sperm washing but also for the subsequent proteomic analysis. It is well known that multistep proteomic workflows may lead to significant sample losses, especially when working with paucicellular samples. In clinical fields, researchers often have to deal with a low protein content in the samples that are available for downstream analysis, thereby hampering comprehensive characterization.

Indeed, spermatozoa are an ideal cell type for such studies, as they can be isolated as a highly purified and relatively homogeneous material source. Nevertheless, it is known that the testicular spermatozoa are functionally immature cells, although morphologically differentiated. They are incapable of progressive movement, and they lack the ability to fertilize an egg [11,12,13]. Human spermatozoa acquire motility and fertilizing potential after leaving the testis, during which sperm cells are in contact with the fluids secreted by the epididymis, prostate, seminal vesicles, and bulbourethral glands, which collectively produce the seminal plasma [14]. Thus, post-testicular functional maturation of the spermatozoa and their motility is attributed to the acquisition of components from the seminal plasma [13,15]. Seminal plasma has not yet received much attention from the proteomics community, but its characterization could provide a future reference for virtually all studies involving human sperm [16]. Moreover, seminal plasma proteins can serve as potential markers of impaired spermatogenesis [17].

Thus, in order to achieve in-depth protein coverage of abnormal semen, a comprehensive analysis of both sperm cells and seminal plasma is necessary [6]. Therefore, the determination of the function, localization, and PPI of specific fertility-related proteins differentially expressed between samples from fertile and infertile men is required [18]. As a rule, the spermatozoa and seminal plasma proteomes are investigated separately for this purpose [19,20]. In turn, this leads to an increase in the cost and timing of the proteomic analysis, which limits its large-scale use in clinical applications.

The purpose of this work was to assess the eligibility of using the ejaculate samples for identifying factors of male fertility status in an express analysis. The human sample size was limited, so our results are preliminary and descriptive.

## 2. Materials and Methods

### 2.1. Biospecimen Collection

This study was approved by the local ethics committee of the Federal State Autonomous Educational Institution of Higher Education, N.I. Pirogov Russian National Research Medical University of the Ministry of Health of the Russian Federation. All participants signed an informed written consent form. The semen samples from healthy normozoospermic fertile men (n = 3) were used for proteomic analysis. Semen samples were collected via masturbation, after 2–3 days of sexual abstinence. Semen volume, sperm motility, and concentration were evaluated according to the WHO 2021 guidelines: sperm parameters were considered normal when the semen volume was 1.4 (1.3–1.5), sperm concentration was 39 (35–40 million) per ejaculate, total sperm motility was 42 (40–43%), and normal forms were 4 (3.9–4%) [3]. For the proteomic analysis, semen samples from all subjects were used after preliminary freezing in liquid nitrogen (−196 °C) and stored at −80 °C for 2 to 3 weeks.

Following thawing on ice, each of the three ejaculates from three donors were divided into two parts. The first one (1/3 of ejaculate) was used wholly for further semen protein solubilization. The second part that made up 2/3 of the ejaculate was used for the spermatozoa washing procedure. Each of these portions were centrifuged at 4 °C (2000× *g*, 10 min) to separate the spermatozoa from the seminal plasma. Spermatozoa were washed three times using PBS (phosphate-buffered saline) buffer, pH 7.4. After each centrifugation, the supernatants were collected and combined from each sample into their own tube (Eppendorf, Macquarie Park, Australia). In total, three samples (ejaculate, seminal plasma, and spermatozoa) were obtained for every donor. For protein identification via LC–MS/MS, the aliquots of ejaculate, seminal plasma, and spermatozoa pellets were processed immediately after being obtained.

### 2.2. Sample Preparation for LC–MS/MS

Aliquots of ejaculate and seminal plasma were mixed with 2% SDS in 100 mM Tris-HCl (pH = 7.4), 120 mM NaCl, 5 mM EDTA, 1% PMSF in a ratio of 1:1 (*v*/*v*) and manually homogenized in a Potter glass tissue homogenizer. The spermatozoa pellet was placed into 20 μL of 100 mM Tris-HCl buffer (pH = 7.4) containing 2% SDS, 120 mM NaCl, 5 mM EDTA, 1% PMSF and manually homogenized in the Potter glass homogenizer with a Teflon pestle. Then, all the samples were sonicated in an ice-cold bath (active time 25 s) and incubated for 30 min at +4 °C on an orbital shaker with a platform rotation of 1000 rpm. After heating at 95 °C for 4 min, they were centrifuged at 14,000× *g* for 20 min (+4 °C). The lysate was collected, and the procedure was repeated, beginning with the sonication step. Protein lysates were pooled and centrifuged at 14,000× *g* for 60 min (+4 °C). Afterwards, the supernatant was used for 1DE-gel concentration procedure [21].

The total protein concentrations of ejaculate, seminal plasma, and spermatozoa extracts were determined via the bicinchoninic acid assay [22], using BSA as a standard on the Agilent 8453 UV–visible spectrophotometer (Agilent, Santa Clara, Ca, USA).

In order to remove SDS, the 1DE-gel concentration procedure in denaturation conditions was then carried out using Protean II xi Cell (Bio-Rad, Hercules, CA, USA). Each SDS-containing protein lysate was mixed with the sample buffer (4% SDS, 100 mM DTT, 10% glycerol), heated at 95 °C for 5 min, and deposited onto 4%T polyacrylamide stacking gel (50 μg of protein per gel lane). Electrophoresis was carried out at 50 V and terminated before the migration of bromophenol blue into the resolving gel (12%T). Protein bands (one per gel lane) were visualized via staining with Coomassie Brilliant Blue R250, and then each of them was completely cut out, moved to a distinct Eppendorf, and destained.

Each gel band was further crushed using a scalpel, and in-gel digestion with trypsin was performed according to the standard procedure [23]. Briefly, each band was incubated in destaining buffer (50% acetonitrile (*v*/*v*) in 100 mM ammonium bicarbonate, pH = 8.9) at 50 °C for 45 to 60 min; destaining was repeated twice. Next, each probe was reduced with 45 mM DTT at 56 °C for 60 min, and then alkylation was performed with 100 mM IAA at room temperature for 15 min (in the dark). After dehydration, each probe was subjected to in-gel proteolysis with trypsin. For this purpose, trypsin solution (25 ng/μL modified trypsin in 50 mM bicarbonate ammonium) was added to each probe, depending upon its relative staining (6.3 ± 2.0 μL), and mixtures were incubated for 1 h at 37 °C. After this, an additional trypsin solution was added to each probe, and the mixtures (gel piece with trypsin) were incubated for 18 h at 37 °C. Then, 15 μL of 0.7% trifluoroacetic acid was added to each gel piece, and the samples were incubated for 2 h at room temperature. The mixture of proteolytic peptides from each gel band was used for LC–MS/MS analysis.

### 2.3. LC–MS/MS Analysis

Separation and identification of the peptides were performed on an Ultimate 3000 nano-flow HPLC system (Dionex, Sunnyvale, CA, USA), connected to the Orbitrap Q Exactive mass-spectrometer (Thermo Scientific, Waltham, MA, USA) equipped with a Nanospray Flex NG ion source (Thermo Scientific, USA) as described earlier [21].

### 2.4. Data Processing

Nineteen LC–MS/MS runs were carried out for ejaculate, seminal plasma and spermatozoa extracts from three donors. The initial RAW files were converted to MGF files with the ProteoWizard MSConvert program [24]. Files were imported into the SearchGUI (v. 4.2.2) platform [25] and searched with the X!Tandem and MS-GF+ search algorithms against the SwissProt human database (v. 22.03.2022, https://www.uniprot.org/proteomes/UP000005640 (accessed on 20 September 2023), FASTA format) with the following search parameters: enzyme trypsin; the maximum number of missed cleavages 1; fixed modification carbamidomethylation of C (in-gel digestion); variable modification oxidation methionine. Parent and fragment ions were searched with tolerances of ±5 ppm and ±0.01 Da, respectively. The PeptideShaker integrator [26] was used to obtain an Excel spreadsheet file with the results.

Results were imported into the Functional Enrichment analysis tool (Funrich v. 3.1.3 [27]) to build Venn diagrams and perform Gene Ontology (GO) enrichment analysis (database v. 12.12.2021, http://geneontology.org/docs/downloads/ (accessed on 20 September 2023)). GO analysis (including the biological process, cell component and molecular function) of ejaculate, seminal plasma and spermatozoa proteins was performed using the g:Profiler ([28], https://biit.cs.ut.ee/gprofiler/, accessed on 20 September 2023). HPA ([29], https://www.proteinatlas.org/, accessed on 20 September 2023) and Metascape ([30], https://metascape.org/, accessed on 20 September 2023) were used to analyze testis-specific proteins of ejaculate, seminal plasma, and spermatozoa specimens. The version of g:Profiler was e109_eg56_p17_1d3191d (database updated on 14.06.2023). The parameters for the enrichment analysis were as follows: a specific organism was chosen *Homo sapiens*, GO analyses (GO biological process (GO:BP)) and GO cellular component (GO:CC) were carried out sequentially.

## 3. Results and Discussion

### 3.1. Comparison of the Ejaculate, Seminal Plasma, and Spermatozoa Proteomes

In recent years, cases of male infertility have increased, therefore, the male reproductive system has received both medical and social attention. As with any other disorder, an in-depth diagnosis of male infertility requires high-tech and sensitive methods. New-generation techniques, such as proteomics, can be clinically used to identify the specific cause and biomarkers of male infertility.

At the same time, the process of sample preparation for proteomic analysis (including choosing a suitable tissue for identifying aberrant cellular mechanisms among infertile males) should not take much time and should be relatively simple and reproducible. Decreasing the biospecimen processing steps may simplify the sample preparation procedure. For diagnostics (at least express analysis) of male infertility using proteomic approaches, the use of ejaculate is useful since several steps of centrifugation during sperm washing can be excluded.

In this study, a shotgun analysis of the ejaculate of healthy normozoospermic fertile men was performed and compared with the results obtained during the proteomic analysis of seminal plasma and spermatozoa divided from the same ejaculate. Sperm samples were characterized by the following parameters: semen volume was ≥1.5 mL, sperm concentration was 70.8 ± 27.4 million/mL, total sperm motility was 68.2 ± 14.2%, and normal sperm morphology was ≥4%. Notably, some indicators were slightly higher than the WHO standard for sperm quality [3].

To characterize semen proteome, we performed SearchGUI processing by joining the technical repeats within each dataset (seven runs for both ejaculate and seminal plasma and six runs for spermatozoa). LC–MS/MS detected a total of 2647 proteins. Among them, 1779, 715, and 2163 proteins were identified in the ejaculate, seminal plasma, and spermatozoa, respectively (Appendix A). In order to avoid false-positive findings, subsequent processing was performed within protein identifications via ≥2 unique, validated peptides. 

The similarities and differences in the protein profiles can be analyzed using the Venn diagram (Figure 1a). There were 472 common proteins in all the datasets, of which 22 were associated with the process of sperm cell development (spermatogenesis), according to the Human Proteome Atlas (HPA, [29], https://www.proteinatlas.org/search/spermatogenesis, accessed on 20 September 2023). There were 206, 37, and 412 unique proteins in the ejaculate, seminal plasma, and spermatozoa, respectively. Among them, there were six, one, and sixteen proteins in the ejaculate, plasma, and cells, respectively, which were associated with spermatogenesis. In particular, in the ejaculate, we found Intragut transport protein 27 (Q9BW83), which is essential for male fertility, spermatogenesis, and the formation of sperm flagella. In spermatozoa, we identified proteins such as Spermatid maturation 1 (Q8N4L4) and the Calcium-binding protein spermatid associated 1 (Q96KC9), which are necessary for proper cytoplasm removal during spermatogenesis and for maintaining the structural integrity of the sperm flagella, respectively.

In total, there were 45 spermatogenesis-associated proteins, accounting for approximately 8% of the total number of proteins involved in this process. With that, 28 (5%) and 38 (7%) proteins were identified in the ejaculate and spermatozoa, respectively. The smallest number of proteins (n = 23) was in the seminal plasma, which accounted for 4% of the total number of spermatogenesis-associated proteins (565 proteins), according to the HPA.

Thus, it can be concluded that, in comparison with each other, none of the biological samples showed significant power as a source for searching for proteins associated with the process of sperm cell development within the male reproductive organs, the testes.

### 3.2. GO Enrichment Analysis

Gene Ontology (GO) is one of the main resources of biological data which can provide information on biological pathways that gene lists generated from proteomic experiment are involved in [31]. Based on the GO annotation of the full genome, it is possible to assess if the group of genes has some coherent functional signal that can be used to formulate biological hypotheses.

GO enrichment analysis was performed using the g:Profiler toolset [28] to identify the statistically enriched cellular components (CC) and biological processes (BP) between all biological samples (ejaculate, seminal plasma, and spermatozoa, Figure 2). Among the most abundant GO terms characteristic of the total ejaculate, seminal plasma, and spermatozoa proteins (472 proteins, Figure 1a), it was possible to distinguish biological processes (GO:BP) such as single fertilization (GO:0007338), spermatid development (GO:0007286), and cell motility (GO:0048870). There were 89 (19%) proteins involved in these biological processes closely related to the developmental process by which male germ line stem cells self-renew or give rise to successive cell types, resulting in the development of a spermatozoa (spermatogenesis, GO:0007283, https://www.ebi.ac.uk/, accessed on 20 September 2023). Among the true abundant GO terms for CC (GO:0005575), the zona pellucida receptor complex (GO:0002199) and the sperm fibrous sheath (GO:0002199) were presented by six and five proteins, respectively. However, this accounts for approximately 71–75% of the corresponding “term size” in both cases. Among the GO terms enriched via g:Profiler in CC, the term “outer dense fiber” (GO:0001520) was revealed. Defects in the outer dense fibers (ODFs) lead to abnormal sperm morphology and infertility. We identified three ODFs, including ODF1 (Q14990) and ODF2 (Q5BJF6), which belong to the major outer dense fiber proteins.

GO enrichment analysis of the unique proteins in the ejaculate (206 proteins, Figure 1a), seminal plasma (37 proteins, Figure 1a), and spermatozoa (412 proteins, Figure 1a) revealed no statistically significant GO terms (neither among GO:BP nor among GO:CC) that were direct descendants or co-occurring terms of GO:0007283 and GO:0005575.

Taken together, our results demonstrate that the ejaculate can be used, apparently, along with spermatozoa and seminal plasma, to search for specific proteins that regulate male reproductive health and are associated with the fertilization process.

### 3.3. Testis-Specific Proteins

The main functions of the testes are the biosynthesis of hormones for the development of male sex characteristics and spermatogenesis. For a sufficient period of time, testicular biopsies served as the main diagnostic approach for men with unexplained (idiopathic) infertility and azoospermia. Nonetheless, the application of testicular biopsies was restricted as an alternative diagnostic and predictive tool for the confirmation of nonobstructive azoospermia in men [32].

The success of choosing a biological sample for diagnosis may be assessed via the number of revealed tissue-specific proteins. However, it would be more reliable to compare tissue-specific proteins and/or whole protein families detected in different specimens. According to the HPA data [29], at least 130 proteins of the testes or male excretory glands (appendage of the testis, duct of the vas deferens, prostate gland, and seminal vesicles) are known to be locally secreted in the male reproductive system. Thus, in our opinion, testicular-specific proteins can be used to assess the quality of human sperm (e.g., for the express proteomic diagnosis of male infertility).

Herein, we identified 139 testis-specific proteins (Appendix A), which is equal to about 12% of all proteins denoted to human male gonads, according to the Human Protein Atlas [29]. Among them, 90 and 133 proteins were detected in the ejaculate and spermatozoa, respectively. A smaller number of revealed testis-specific proteins (n = 34) were observed in the seminal plasma. The Venn diagram using FunRich [27] presented both the differences and similarities between the testis-specific protein datasets corresponding to the ejaculate, plasma, and cells (Figure 1b). Testis-specific proteins shared between ejaculate, seminal plasma, and spermatozoa included 34 identifications, and they all contained proteins identified in seminal plasma in full force. Furthermore, 63% of the proteins of spermatozoa can be identified in ejaculate.

It has been suggested that the expression of genes encoding proteins related to fertilization may serve as markers for predicting male fertility [33]. Table 1 lists some testis-specific proteins that are considered candidates towards a marker panel for male fertility impairment [34,35]. Among the proteins secreted in the male reproductive system, we identified acrosin (P10323, ACR), which is important in the merger between sperm and egg. Acrosin breaks down the zona pellucida, enabling the sperm to reach the glycoprotein layer of the oocyte. Total acrosin activity may be considered a sensitive biochemical marker for the clinical evaluation of unexplained infertility in males [36]. In addition to ACR, the zona pellucida binding protein (Q9BS86, ZPBP) and A-kinase anchor protein 4 (Q5JQC9, AKAP4) were also highlighted due to their role in male reproductive tissues, association with infertility phenotypes, and participation in specific biological functions in spermatozoa [35]. AKAP4 plays a crucial role in maintaining the integrity of sperm fibrous sheaths and normal sperm motility. After Akap4 was knocked out, the spermatozoa were found with shortened, bent, coiled, and pronged flagella and reduced diameters in the principal piece in mice. As a result, the sperm motility decreased, and the male was infertile [37].

As can be seen from Table 1, the main parts of proteins associated with male infertility (73%) were found in all of the studied subtypes of biological samples (ejaculate, seminal plasma, and spermatozoa). Three proteins (PRM2, DYNLT1, and EQTN) were identified both in ejaculate and cells, which therefore confirms our observation about the possibility of using ejaculate in clinical assays, contributing to the development of methods for the diagnosis and treatment of infertility. We identified lymphocyte antigen 6K (Q17RY6, LY6K) in the ejaculate only. This protein is specifically expressed in testicular germ cells, where it plays a role in sperm cell migration [38]. Later experiments proved that although LY6K-deficient mice can produce morphologically intact spermatozoa, they display an infertile phenotype due to the inability of spermatozoa to migrate into the oviduct [39]. Thus, our data indicated that the ejaculate, along with spermatozoa (or even instead of), may be suitable diagnostic specimens for assessing male fertility.

**Table 1 biomedicines-12-00049-t001:** The list of some testis-specific proteins associated with male infertility and identified in human ejaculate, seminal plasma, and spermatozoa samples.

Accession	Gene	Protein Name	Validated Unique Peptides			Spectrum Counting NSAF			Function/Disorder
			Ejaculate	Plasma	Spermatozoa	Ejaculate	Plasma	Spermatozoa	
Q5JQC9	*AKAP4*	A-kinase anchor protein 4	53	33	58	1.322	0.655	2.928	Major structural component of sperm fibrous sheath. Plays a role in sperm motility/male infertility [35,37,40]
O14556	*GAPDHS*	Glyceraldehyde-3-phosphate dehydrogenase, testis-specific	13	10	32	1.264	0.747	3.647	Required for sperm motility and male fertility/azoospermia [41]
Q9BS86	*ZPBP*	Zona pellucida-binding protein 1	18	12	20	0.743	0.610	1.145	Plays a role in acrosome compaction and sperm morphogenesis. Is implicated in sperm-oocyte interaction during fertilization/male infertility [35]
Q9BY14	*TEX101*	Testis-expressed protein 101	9	10	10	0.532	0.597	0.422	Plays a role in fertilization by controlling binding of sperm to zona pellucida and migration of spermatozoa into the oviduct/azoospermia [38,42,43]
P54107	*CRISP1*	Cysteine-rich secretory protein 1	29	30	15	3.781	4.586	1.375	May have a role in sperm-egg fusion and maturation [44]
P16562	*CRISP2*	Cysteine-rich secretory protein 2	11	30	8	0.49	0.383	0.474	May regulate some ion channels’ activity and thereby regulate calcium fluxes during sperm capacitation [45]
P12273	*PIP*	Prolactin-inducible protein	65	62	47	28.654	32.801	14.785	Regulation of immune system process [46]
P10323	*ACR*	Acrosin	9	8	12	0.489	0.489	0.762	The major protease of mammalian spermatozoa/globozoospermia, spermatogenic failure 9 [36]
P04554	*PRM2*	Protamine-2	6	nd	13	0.338	nd	2.044	Protamines substitute for histones in the chromatin of sperm during the haploid phase of spermatogenesis/male infertility, azoospermia [47]
P26436	*ACRV1*	Acrosomal protein SP-10	4	3	7	0.319	0.139	1.444	Spermatogenesis (by similarity, [48])
P54652	*HSPA2*	Heat shock-related 70 kDa protein	41	22	26	0.704	0.541	0.743	Plays a role in spermatogenesis. In association with SHCBP1L may participate in the maintenance of spindle integrity during meiosis in male germ cells/varicocele [49,50]
P63172	*DYNLT1*	Dynein light chain Tctex-type 1	2	nd	4	0.16	nd	0.311	Acts as a motor for the intracellular retrograde motility of vesicles and organelles along microtubules [51]
Q14990	*ODF1*	Outer dense fiber of sperm tails 1	7	10	43	0.235	0.241	3.341	Component of the outer dense fibers (ODF) of spermatozoa. ODF are filamentous structures located on the outside of the axoneme in the midpiece and principal piece of the mammalian sperm tail and may help to maintain the passive elastic structures and elastic recoil of the sperm tail/spermatogenic failure 9 [52]
Q9NQ60	*EQTN*	Equatorin	1	nd	2	0.029	nd	0.033	Acrosomal membrane-anchored protein involved in the process of fertilization and in acrosome biogenesis [53]
Q17RY6	*LY6K*	Lymphocyte antigen 6K	2	nd	nd	0.034	nd	nd	Required for sperm migration into the oviduct and male fertility through controlling binding of sperm to zona pellucida [38,39]

### 3.4. Functional Enrichment Analysis of Testis-Specific Proteins

To gain a more meaningful understanding of ejaculate employment for male infertility screening, the association between testis-specific proteins identified in three datasets (ejaculate, seminal plasma, and spermatozoa) was analyzed using functional annotation databases in Metascape with the following ontology categories: PaGenBase [54] and DisGeNET [55]. Results were visualized via heatmaps that represent two-dimensional tables to depict the relationship between analyzed datasets (spermatozoa, seminal plasma, and ejaculate) and related biological categories from PaGenBase and DisGeNET knowledgebases (Figure 3a,b). Color intensities in heatmaps represented the *p*-values of the corresponding biological categories enrichment (tissue/cell type or disease) within each dataset.

As can be seen from Figure 3a, only two terms were abundant in the Pattern Gene Database (PaGenBase): testis interstitial (tissue-specific) and testis germ cells (cell-specific). Unsurprisingly, there was a rather high overlap between testis-specific proteins related to different tissue types. However, spermatozoa and ejaculate demonstrated high enrichment to a greater extent.

Male fertility is maintained through intricate cellular and molecular interactions, and it is dependent on the continual production of sperm, starting at puberty and continuing throughout adult life. Sertoli cells have long been assumed to be the major cellular players in testis organogenesis and spermatogenesis. However, the interstitial tissue of the testis has been shown to play a prominent role, which consists of mechanical support to the seminiferous tubules and blood vessels, the production of testosterone by interstitial cells (Leydig cells), participation in the sustentacular cell barrier, and regulation of the sustentacular cell. Interstitial cells are critical links between the immune system, the reproductive system, and overall health [56]. The thickening of the tubular basement membrane and interstitial blood vessel wall with luminal narrowing, as well as the proliferation of Leydig cells and increased deposition of interstitial fibrous tissue, can lead to a decrease in the concentration of spermatozoa and spermatogenesis. In addition, the basis of male fertility also forms the germ line stem cells in the mammalian testis. Germ cells have promising clinical applications for patients who undergo sterilizing treatments for cancer or targeted correction of genetic defects in testicular somatic cells. In these cases, spermatogenesis restoration can be achieved via germ cell transplantation, in which donor testicular cells are transferred into the testes of infertile men [57].

Disease enrichment analysis using the DisGeNET platform demonstrated that the testis-specific proteins of ejaculate, seminal plasma, and spermatozoa were associated with male infertility, astenozoospermia, and malignant neoplasm of the testis (Figure 3b). Only the ejaculate and spermatozoa contained the teratozoospermia-associated genes/proteins. These observations found reflection in the clustering results: spermatozoa and ejaculate proteomes were grouped in one and the same cluster. Thus, this indicates a possible usage for ejaculate along with or even instead of spermatozoa to distinguish various forms of male infertility (in our case, from astenozoospermia and teratozoospermia to cancer). As can be seen from Figure 3b, azoospermia disease-associated genes/proteins were found only in spermatozoa, however, it should be noted that the enrichment was rather low. Moreover, we could detect in the ejaculate, as well as in plasma and cells, testis-expressed protein 101 (Q9BY14, TEX101), the seminal plasma level of which is associated with the subtypes (obstructive and nonobstructive) of azoospermia [42,43]. Furthermore, the clinical assay for TEX101 has the potential to replace most of the diagnostic testicular biopsies and facilitate the prediction of outcomes of sperm retrieval procedures, thus increasing the reliability and success of assisted reproduction techniques [42].

Using Metascape analysis, the PPI networks of the testis-specific proteins were constructed, and the molecular complex detection (MCODE) algorithm [58] was applied to the resultant networks to identify the tightly connected network cores. There were three PPI modules, which included sixteen genes/proteins, and six of them, AKAP4 (Q5JQC9), ROPN1 (Q9HAT0), AKAP3 (O75969), ROPNL1 (Q96C74), SPA17 (Q15506), and ROPN1B (Q9BZX4), were associated with cilium movement (Figure 4a). Among the top list of enriched terms, we found two more major GO:BP clusters: microtubule-based movement (six proteins, Figure 4b) and sperm motility (four proteins, Figure 4c). In addition, the KEGG pathway, hsa01200 (carbon metabolism), also referred to the three best-scoring terms by *p*-value (lg(*p*) = −6.7).

As is observable from Figure 4a–c, seven proteins (AKAP4, AKAP3, SPA17, ROPN1B, CAPZA3, ACTRT2, and TKTL2) belonging to the obtained PPI modules were found in all three specimens. Eight proteins (ROPN1, ROPN1L, ACTL9, ACTRT3, BANF2, ACTL7A, PDHA2, and LDHAL6B) were detected both in the ejaculate and spermatozoa, and TKTL1 was revealed only in the cells.

In summary, by means of gene–gene network reconstruction analysis, we found 16 genes/proteins associated with spermatogenesis (GO:0007283) and male gamete generation (GO:0048232). Here, we focused the discussion on four proteins: Q9HAT0 (Rhophilin-associated tail protein, ROPN1), Q8TC94 (Actin-like protein 9, ACTL9), Q5JQC9 (A-kinase anchoring protein 4, AKAP4), and Q15506 (Sperm autoantigenic protein 17, SPA17). The protein Q9HAT0, encoded by the gene *ROPN1*, is important for male fertility, and it is related to the PKA-dependent signaling processes required for spermatozoa capacitation. With the Ropporin-1-like protein (ROPN1L), Q9HAT0 is involved in fibrous sheath integrity and sperm motility. Diseases associated with ROPN1 include varicocele, where veins enlarge inside the scrotum (the pouch of skin holding testicles) [59]. This causes the testicles to overheat, which can reduce the production and function of sperm and thus affect fertility. Testis-specific Q8TC94 plays an important role in the fusion of proacrosomal vesicles and perinuclear theca formation. Diseases associated with ACTL9 include spermatogenic failure 53 and non-syndromic male infertility due to sperm motility disorder. Spermatogenic failure 53 is characterized by oocyte fertilization failure due to a lack of oocyte activation, which is associated with ultrastructural abnormalities of the sperm head [60]. Similar to Q9HAT0, A-kinase anchoring protein 4 (Q5JQC9, AKAP4) is the major structural component of sperm fibrous sheaths, and it plays a role in sperm motility. Diseases associated with AKAP4 include non-syndromic male infertility due to asthenozoospermia [40]. The protein SPA17 was initially characterized by its involvement in the binding of sperm to the zona pellucida of the oocyte. Through database cross-checking, we found that it is also involved in additional cell–cell adhesion functions such as immune cell migration and metastasis. Recently, SPA17 has gained attention because of its involvement in cryptorchidism, unilateral or bilateral, and testicular germ cell cancer [61]. Therefore, comparing the ejaculate and spermatozoa proteomes could provide a panel of semen proteins with the potential to evaluate sperm quality.

## 4. Conclusions

Proteomics includes an array of techniques aimed at analyzing many proteins in parallel and bearing the potential for improving diagnostic, prognostic, and predictive tests, especially in the context of precision medicine. The utilization of protein panels, rather than individual protein biomarkers, can provide a more comprehensive representation of human physiology, which can potentially improve the diagnosis and prognosis of a human pathophysiological state. The method for perfect screening, as well as diagnosing any human pathology, including male infertility, must be easily reproducible, sensitive, scalable, cost-effective, and, most importantly, effective [62]. The choice of a suitable biological sample for proteomic analysis is also of significant importance. In biomedicine, proteomics can be applied to characterize accessible biological fluids, tissues, and cells to provide information about health and case features [63]. Furthermore, spermatozoa or seminal plasma are suitable samples for proteomic analysis of the male reproductive system [19]. We believe that ejaculate could serve as one of the analytic specimens to determine fertility, is obtainable with relative ease, and contains many proteins. Moreover, the testis-specific proteins could form the basis of a protein panel for detecting disorders associated with male infertility.

Herein, we conducted a comparative analysis of ejaculate, spermatozoa, and seminal plasma proteomes for the possibility of using the ejaculate to assess sperm quality and to detect spermatogenesis abnormalities. The focus was set on the examination of testis-specific proteins that participate in the production of sperm and are important for the male fertility score. Utilizing the bioinformatics pipeline for data analysis, it was revealed that the proteins secreted by the male reproductive system of ejaculate were associated with the same human diseases and types of tissues/cells as the proteins of spermatozoa and seminal plasma. The protein–protein interaction networks of the ejaculate, spermatozoa, and plasma testis-specific proteins were constructed, and by means of network reconstruction analysis, we found 16 genes/proteins that were associated with spermatogenesis (GO:0007283) and male gamete generation (GO:0048232). Furthermore, almost all these proteins were detected in both the ejaculate and spermatozoa.

In addition, the preliminary results of the proteomics analysis of oligozoospermia patients’ ejaculates were also obtained. Compared to the normozoospermic ejaculate identifications, at oligozoospermia, we revealed a decrease in the number of spermatogenesis-associated proteins. For example, in these samples, we could not detect outer dense fiber proteins 1 and 2, sperm acrosome membrane-associated protein 1, voltage-dependent anion-selective channel protein 3, and sperm protein associated with the nucleus on the X chromosome B1. While further investigations into various pathological sperm conditions need to be conducted, we believe that a proteomic analysis based on testis-specific proteins of ejaculate along with spermatozoa (or even instead of) can be useful for distinguishing fertile, sub-fertile, and infertile men. Moreover, it can help improve the diagnosis, define particular causes, and predict the outcomes of infertility. In addition, large-scale proteomics may reveal specific protein patterns (for example, differently expressed/housekeeping/testis-specific proteins or their combinations), which, in conjunction with bioinformatics analysis, can help reveal the perturbed biological pathways connected to male infertility. This, in turn, may facilitate the accumulation of evidence concerning disease etiology at the proteomic level, which is of particular importance since spermatozoa lack transcription and translation. Therefore, we suppose that ejaculate proteomic patterns can enhance our understanding of male reproductive physiology and treatment efficiency, including target-based drug discovery.

## Figures and Tables

**Figure 1 biomedicines-12-00049-f001:**
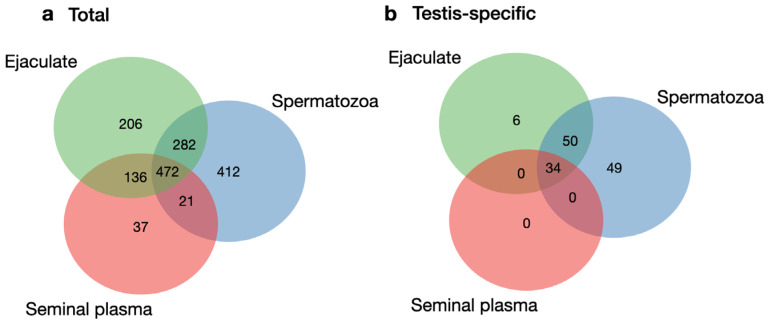
Unique and shared proteins identified via ≥2 validated unique peptides in ejaculate, seminal plasma, and spermatozoa. Venn diagrams presenting (**a**) total numbers of proteins (**b**) testis-specific proteins (according to the Human Protein Atlas database [29], v. 20062022, https://www.proteinatlas.org/humanproteome/tissue/testis, accessed on 20 September 2023).

**Figure 2 biomedicines-12-00049-f002:**
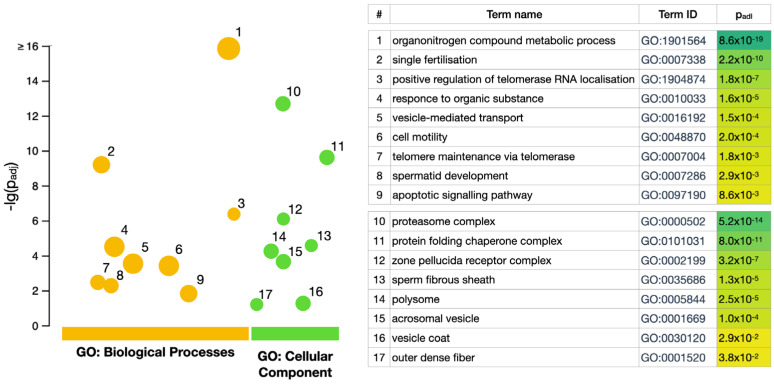
Gene set enrichment analysis performed using g:Profiler [28] (https://biit.cs.ut.ee/gprofiler/gost, accessed on 20 September 2023). The circle size represents the number of genes corresponding to certain term of biological processes (GO:BP) or cellular components (GO:CC).

**Figure 3 biomedicines-12-00049-f003:**
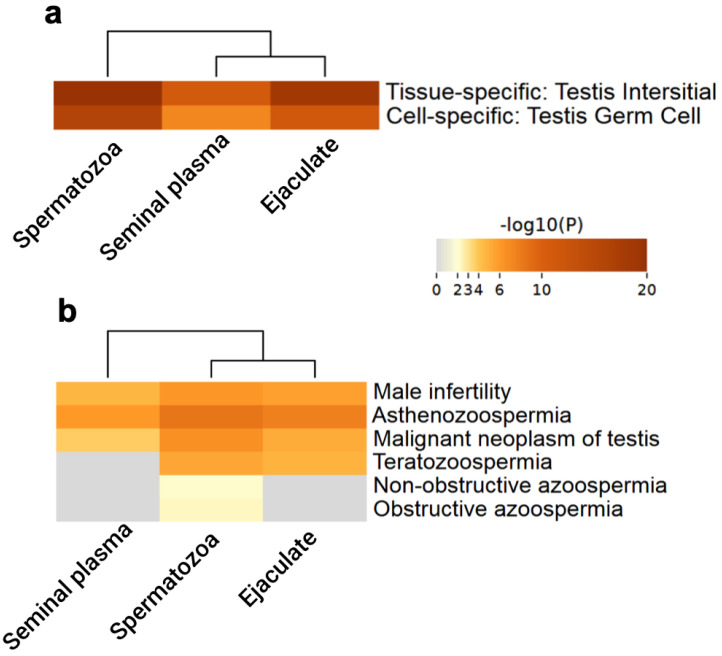
Summary of functional enrichment analysis of testis-specific proteins (**a**) in Pattern Gene Database (PaGenBase, v. 1.0), which involves kinds of time/tissue-specific conditions from the aspects of tissue/cell type, development/differentiation and so on; and (**b**) on a discovery platform containing one of the largest publicly available collections of genes and variants associated to human diseases DisGeNET (v. 7.0). Heatmaps of enriched terms across input gene lists with the discrete color scale to represent statistical significance was constructed with the Metascape web tool (https://metascape.org/, accessed on 20 September 2023).

**Figure 4 biomedicines-12-00049-f004:**
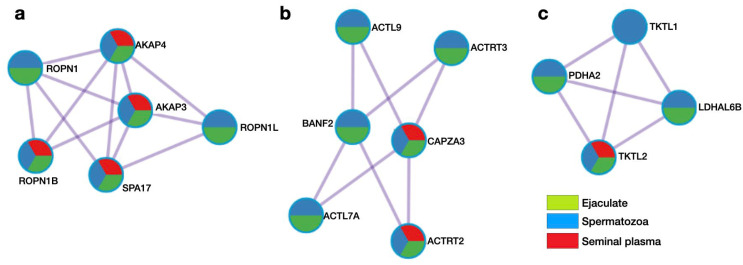
Molecular complex detection (MCODE) analysis of protein–protein interaction (PPI) network of testis-specific proteins identified in ejaculate (green color), spermatozoa (blue color) and seminal plasma (red color). (**a**) Cilium movement (GO:0003341, lg(*p*) = −10.5); (**b**) microtubule-based movement (GO:0007018, lg(*p*) = −8.7); (**c**) sperm motility (GO:0097722, lg(*p*) = −8.5).

## Data Availability

The data presented in this study have been deposited in the ProteomeXchange Consortium via the PRIDE partner repository, and are openly available under accession PXD046848.

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
