# Peer review of "The Application of Ejaculate-Based Shotgun Proteomics for Male Infertility Screening"

_biomedicines, 2023, doi:10.3390/biomedicines12010049_

Round 1

Reviewer 1 Report

Comments and Suggestions for Authors

The topic is interesting and the research conducted by the authors is contemporary by means of experimental methods. However, the results are mainly fundamental and of less practical importance. Regarding the manuscript, I have the following remarks:

1. In the Materials and Methods section, it is not clear by which WHO criteria was the initial spermatozoa evaluation. They have written 2010, but in the Reference list they have cited 2000. 

2. In the same paragraph they write that “….sperm concentration was 70.8 +- 27.4 mln….”, but that is not a WHO criterion. If it is their result, it should be cited in the appropriate section of the manuscript.

3. They have frozen the spermatozoa in liquid nitrogen, but have kept them at -80oC. The temperature of the liquid nitrogen is -196oC. 

4. The conclusions state that the proteomics analysis can be used for distinguishing fertile, subfertile and infertile patients. I definitively think that to be able to draw such a conclusion, the research should be conducted not only on normozoospermic ejaculates, but also astenozoospermic, teratozoopsermic, etc. should be compared. Also, the number of investigated ejaculates is too low - only 3.

Comments on the Quality of English Language

The English language needs improvement, there are a lot of repetitions, which can be avoided, in order to make the paper more readable. Some sentences are unclear and ambiguous.

Author Response

We express our deep gratitude to the Reviewer for the attention to our manuscript and for valuable comments. The authors agree with the comments and recommendations of the Reviewer and have taken them into account while preparing a revised version of the manuscript.

Point 1: In the Materials and Methods section, it is not clear by which WHO criteria was the initial spermatozoa evaluation. They have written 2010, but in the Reference list they have cited 2000.

Response 1: Thank you for pointing this out; authors apologize for the technical error. The appropriate correction was made (Ref 3).

Point 2: In the same paragraph they write that “….sperm concentration was 70.8 +- 27.4 mln….”, but that is not a WHO criterion. If it is their result, it should be cited in the appropriate section of the manuscript.

Response 2: We agree that the provided criterion is not WHO standard. This is our result, and according to your suggestion, we cited it in the appropriate manuscript section (lines 210-214). Additionally, the 2021 WHO guidelines criteria were added to the Materials and Methods section (lines 112-116).

Point 3: They have frozen the spermatozoa in liquid nitrogen, but have kept them at -80oC. The temperature of the liquid nitrogen is -196oC.

Response 3: Thank you for pointing this out. We tried to describe it in a clear way: “…semen samples from all subjects were used after preliminary freezing in liquid nitrogen (-196°C) and storage at - 80°C during 2-3 weeks” (lines 116-118).

Point 4: The conclusions state that the proteomics analysis can be used for distinguishing fertile, subfertile and infertile patients. I definitively think that to be able to draw such a conclusion, the research should be conducted not only on normozoospermic ejaculates, but also astenozoospermic, teratozoopsermic, etc. should be compared. Also, the number of investigated ejaculates is too low - only 3.

Response 4: We agree. However, since we had a considerable number of technical repeats (from 5 to 7) for each fraction of biological samples, we believe that the preliminary conclusions could be made. Nonetheless, the further validation is needed indeed (not only on normozoospermic ejaculates, but also astenozoospermic, teratozoopsermic, etc. should be compared). We have begun the comparative analysis of oligozoospermia and normozoospermic ejaculates; the pilot results were included as a demo of the possible ejaculate usage for male infertility screening (Conclusions section, lines 473-482).

Comments on the Quality of English Language

According to the Reviewer’s recommendation, we have submitted our manuscript to the MDPI English editing.

In conclusion, we would like once again to thank Reviewer for the throughout review and valuable comments.

Reviewer 2 Report

Comments and Suggestions for Authors

In this work, the authors report the “Application of Ejaculate-Based Shotgun Proteomics for Male Infertility Screening”. The study has interesting results and discussion and is presented well. However, the authors requested to address the following minor comments.

Some critical comments and suggestions

1.      What is GO? Please add an explanation when using it in the beginning.

2.      Table 1. What is ID? Further, the table should be updated with valid references.

3.      Fig. 3. Should be explained clearly.

4.      If it would be better it explained treatment strategies.

Comments on the Quality of English Language

 Minor editing of English language required

Author Response

We express our gratitude to the Reviewer; your comments were extremely helpful. The authors agree with the comments and recommendations from Reviewer and have taken them into account while preparing a revised version of the manuscript.

Point 1: What is GO? Please add an explanation when using it in the beginning.

Response 1: The explanation was added to the text of the manuscript (lines 250-254).

Point 2: Table 1. What is ID? Further, the table should be updated with valid references.

Response 2: We had meant that ID is the serial number to navigate in the Table. In the revised manuscript version, this column was removed, as we found that the numeration of lines is rarely used by other authors. According to your recommendation, Table 1 was updated with valid references for each protein.

Point 3: Fig. 3. Should be explained clearly.

Response 3: Thank you very much for your valuable comment. We have tried to expand the discussion of the results presented in Figure 3 (lines 344-349, 359-363, and 384-390).

Point 4: If it would be better it explained treatment strategies.

Response 4: The authors agree with the Reviewer. The description of the possible treatment strategy was added to the revised version of the manuscript (lines 482-490).

Comments on the Quality of English Language

According to the Reviewer’s recommendation, we have submitted our manuscript to the MDPI English editing.

Thank you once again for your insightful comments.
